# Where Was Past Low-Entropy?

**DOI:** 10.3390/e21050466

**Published:** 2019-05-04

**Authors:** Carlo Rovelli

**Affiliations:** Centre de Physique Théorique (CPT), Aix-Marseille Université, Université de Toulon, CNRS, F-13288 Marseille, France; rovelli@cpt.univ-mrs.fr

**Keywords:** arrow of time, entropy in cosmology, past hypothesis

## Abstract

Where was the past low-entropy of the early universe located? Contrary to some popular answers, I argue that that the dominant source of low-entropy is the fact that a single degree of freedom, the scale factor, was not at equilibrium. I also discuss possible interpretations of the “improbability” of this early low-entropy.

## 1. Introduction

The first part of this paper traces the source of physical irreversibility, in order to identify where precisely low-entropy was located in the early universe. The issue stems from the fact that the standard model of cosmology [1] assumes matter to be at equilibrium in the early universe—hence at maximal entropy—in apparent tension with the past low-entropy needed to understand the ubiquitous irreversibility of macroscopic phenomena [2,3,4]. I argue against some common answers that overplay the role of gravity (see also [5]), and suggest that the dominant source of low-entropy that feeds the observed irreversible behaviour of the universe is a single degree of freedom, the scale factor, which is (in a precise technical sense specified below) out of equilibrium. A key observation on which this first part of the paper is based derives from [6].

The second part of this paper is speculative and open ended. It discusses the remaining open question regarding irreversibility: can we make sense of the “improbability” or “atypicality” of this early low-entropy? I discuss tentative solutions, including the possibility of a perspectival interpretation of entropy [7].

## 2. Part I. Where?

### 2.1. A Simple Model, First Version

Let me start by recalling some well-established facts in thermodynamics. These are illustrated by a simple model that captures some salient aspects of the thermal history of the universe.

Consider an ideal gas in a thermally isolated container whose volume *V* can be modified with a piston, moved by an external force. Assume that the volume is increased from the initial value *V* at some negative time $t_{0}=-t_{f\;in}$, to a maximum value $V_{max}$ at t=0 and then symmetrically decreased back to *V* at a positive time $t_{f\;in}$, so that the volume evolves in time respecting V(t)=V(−t) as in Figure 1. Assume that at time $t_{0}$ the gas was in thermal equilibrium at temperature *T*. Does the entropy of the gas increase in the process?

The answer depends on the rapidity of the change of the volume. If this is much slower than the thermalization time of the gas, the gas remains constantly close to an equilibrium state. From V(t)=V(−t) and the equilibrium relation pV=constant it follows that the pressure *p* satisfies p(t)=p(−t), and therefore the work W=∫pdV done on the gas during the compression is equal to the work extracted from the gas during the expansion. Hence the net exchange of energy is zero. At $t_{f\;in}$ the gas will therefore have the same temperature as the initial one, hence the same entropy, and there is no increase of entropy in the process. The process is reversible.

But if the volume changes fast enough, the process is irreversible and entropy increases: the temperature at $t_{f\;in}$ is higher than at $t_{0}$. Mechanically, this is easily understood: during a fast expansion, there are less molecules that hit the piston than during a fast compression. In the limiting case where the speed of the piston is much faster than molecular speeds, the pressure during expansion is null, hence no work is extracted during the expansion, while work is done in the compression. Since the work on the gas during compression is larger than the work extracted during expansion, energy is put into the gas, and the final temperature is higher than the initial one. The process is irreversible. During the process, the gas finds itself away from equilibrium.

The moral is that a system initially at equilibrium in a space that expands faster than its thermalization time goes out of equilibrium and generates irreversibility.

Below I argue that this model reproduces a key aspect of what has happened to the universe in which we live: its rapid cosmological expansion has driven its matter content out of equilibrium and this is the dominant source of the present irreversibility in the cosmos. Before doing so, however, we need to refine the model.

### 2.2. A Simple Model, Second Version

The model discussed above includes an external force that drives the piston. There are no external forces in the universe: all degrees of freedom interact with one another and belong to the same coupled dynamics.

Among these degrees of freedom is the cosmological phase factor a(t), which is the dynamical variable that governs the volume of each co-moving region of space in a Friedman cosmology
(1)$$ds^{2}=-dt^{2}+a^{2}\left(t\right)\;d{\vec{x}}^{2},$$
where $d{\vec{x}}^{2}$ is the metric of a homogenous three-dimensional space. The single degree of freedom a(t) acts very much like the volume V(t) of the above model, but it is not driven by external forces: it is driven by a dynamical interaction with the other degrees of freedom of the universe. To mimic this aspect of the thermal history of the universe, let us modify the model and get rid of the external force.

To this aim, consider again the gas in a volume with a piston, but assume now that the piston is attached to a spring, and can exchange energy with the gas. See Figure 2.

Let us denote X(t) the position of the piston at time *t*. With appropriate parameters for the dynamics of the piston-spring system and appropriate initial conditions $X\left(t_{0}\right),\dot{X}\left(t_{0}\right)$, the variation of the volume in time between $t_{0}$ and $t_{fin}$ can be set to be similar to the previous case.

If we disregard dissipation, the piston oscillates indefinitely. But since the temperature and the entropy of the gas increase at each oscillation, there is dissipation. The force F=∫pdA that the gas exerts on the piston during expansion is lower than during compression. Therefore the piston–spring system gradually transfers its (kinetic and potential) energy to the gas. The temperature and the entropy of the gas raise. The process is irreversible. The source of irreversibility is now transparent: since there is no external force, irreversibility is simply conventional, statistical redistribution of energy between the gas and the piston–spring system.

In fact, the piston continues to oscillate, but at each oscillation the temperature of the gas raises at the expense of the energy in the piston-spring system. The oscillations slowly reduce in amplitude until the piston sets to its equilibrium position $X_{o}$. More precisely, the piston ends up fluctuating thermally around this position, since it is coupled to the hot gas. At this point thermal equilibrium is achieved, the temperature of the gas stops increasing, and the entropy has reached its maximum. The source of irreversibility in this process is clear: although the gas was in equilibrium by itself, the system of gas+piston–spring was far from equilibrium. There was far more energy in the piston–spring system than its proper equilibrium share prescribed by energy equipartition. Standard entropy growth dissipates irreversibly this energy into the gas as heat.

To start with, the gas was in thermal equilibrium by itself, but the system as a whole was badly out of equilibrium, because of the energy unbalance between the gas and the piston. This is the origin of the later irreversibility. The answer to the questions “What was out of equilibrium? Where was the low entropy?” is clearly: in the piston–spring system, namely in the single variable X(t). A single variable far away from its equilibrium value is sufficient to determine the initial low-entropy and produce irreversibility, because it can hold a big share of free energy. Microscopically: it can hold energy from the rest of the variables, much decreasing their available phase space. That is the initial source of low entropy.

I argue below that this model reproduces the key features of the thermodynamical history of our universe that are responsible for the observable irreversibility. The state of the universe shortly after the Big Bang was badly out of equilibrium because one single degree of freedom, the scale factor a(t), was. The full initial low-entropy of the universe is entirely stored in the out-of-equilibrium value of a single variable. In the following two sections, I make this claim concrete, showing how this process was realized in the real universe.

### 2.3. Metastable States and the Thermal History of the Universe

To understand the thermal history of the universe, the notion of metastable, or quasi-equilibrium state, and the related notion of channel for entropy-increase, are essential.

A pile of wood in a room full of air is thermodynamically quite stable: it can remain for many years as it is. But the basic elements forming it are not at the maximum possible value of their entropy. Far from that. This is obvious from the fact that if we ignite a fire, the wood burns. Burning is a violently irreversible phenomenon that dramatically increases entropy. After burning, the content of the room is reduced to ashes, smoke and vapour, which form a state of the ingredients in the room much higher-entropy than the initial state formed by wood and air. Therefore the constituents forming the pile of wood in the room full of air are in a remarkably stable configuration and yet far from their maximum entropy state. They are in a metastable, or quasi-equilibrium state.

The reason metastable states exist is that there can be obstructions in the phase space of a system, that do not allow the system to easily explore the full region of phase space that is in principle accessible on energetic grounds. The system remains trapped in a relatively small region of phase space for a very long time. In macroscopic terms, the scale of its thermalization time can be very long: longer than the observation time. This may change if some dynamical event allows the system to overcome the obstruction, thus opening a channel through which the system can exit the phase-space region where it was trapped, and move out to a larger region of its phase space. In the example above, the channel is represented by the combustion process and is opened by the ignition of the fire.

Metastable states are ubiquitous and represent vast storages of low entropy from which irreversible phenomena are fuelled.

The most common metastable systems in the universe are the large clouds of hydrogen. The reason they are metastable is that the protons forming them can fuse into helium, and since fusion is an irreversible process, helium is a much higher entropy state of its hadronic constituents than hydrogen. Since there are potential barriers for protons to fuse into helium, which make spontaneous fusion highly improbable, hydrogen is (meta-)stable. But there are processes that can overcome these potential barriers. A large hydrogen cloud has also a slow gravitational instability that makes it progressively clump [8], emitting heat but also increasing pressure and temperature (gravitational systems typically have negative heat capacity [9]) at its center, until the potential barrier preventing hydrogen to fuse into helium becomes insufficient. Hydrogen starts burning, further increasing temperature, and a channel for rapid increase of entropy is open: a star is born.

A star like the sun is a strongly irreversible process. It produces vast amounts of photons full of free energy, that impact the Earth and fuel a huge amount of irreversible phenomena on the Earth’s surface, including the entire biosphere. Hence the entire irreversibility of life can be traced to the low entropy of the initial metastable hydrogen clouds.

How could the the protons in the hydrogen clouds be in a lower-than-maximal-entropy state, if the matter content of the early universe was in thermal equilibrium, namely a maximal entropy state, according to standard cosmology [1]?

The answer is the fact that the expansion of the universe has been too fast for equilibrium to keep up in the matter component of the universe, in analogy with the rapid expansion of the gas in the model discussed above.

The key event in the history of the universe in this regard is the freezing out of the reactions $n+e^{+}↔{\bar{\nu }}_{e}+p$ and $n\;+\nu_{e}↔p+e^{-}$ that happened around the “freeze out” temperature T=0.7 MeV (about one second after the Big Bang). Before this event, protons and neutrons where in thermal equilibrium. From this time on, the temperature is too low to allow thermalization between the two species. Some neutrons had the time to decay to protons, most fused into helium, fixing the primordial helium to hydrogen ratio, which is still observable in the present universe. This helium to hydrogen ratio represents a metastable configuration, analog to the pile of wood of the example above, because hydrogen can ignite and fuse into helium, increasing entropy. This is the low-entropy fuelling the majority of the irreversible phenomena we see, including life.

The thermal history of the universe is therefore analogous to the rapid expansion of the gas in the model discussed above. As in that model, an initial equilibrium system undergoes a rapid volume expansion, too fast for thermalization to keep up, and this generates irreversibility.

Since the cosmological scale factor a(t) is not manoeuvred by an external force from outside the universe, but is rather a dynamical variable interacting with the rest, the proper idealized model or the universe thermal history is the second version of the gas with the piston discussed above. As in that model, the initial state was not at equilibrium because while matter was so, there was a (single) degree of freedom, the cosmological scale factor a(t), out of equilibrium, in the following sense.

For a degree of freedom *a* part of a large system, we say that the value $\left(a_{o},{\dot{a}}_{o}\right)$ of *a* and its time derivative $\dot{a}$, is an equilibrium value, if the number $N(a,\dot{a})$ of microstates of the large system having the values of *a* and $\dot{a}$ is maximized by $a=a_{o},\;\dot{a}={\dot{a}}_{o}$; hence
(2)$${\left.\frac{\partial N(a,\dot{a})}{\partial a}\right|}_{a=a_{o},\dot{a}={\dot{a}}_{o}}={\left.\frac{\partial N(a,\dot{a})}{\partial \dot{a}}\right|}_{a=a_{o},\dot{a}={\dot{a}}_{o}}=0.$$

If this is not the case, we say that the degree of freedom is out of equilibrium. Notice that we say that a variable is out of equilibrium also if there are no equilibrium values at all (which might be the case for the scale factor of the universe).

Consider for simplicity a spatially flat homogeneous cosmological model with vanishing cosmological constant. For a given matter content, the Friedman equation,
(3)$$\frac{{\dot{a}}^{2}}{a^{2}}=\frac{8\pi G}{3}\;\rho ,$$
determines trajectories in the $\left(a,\dot{a}\right)$ plane. Fixing a value of *a*, a trajectory with higher ρ and higher $\dot{a}$ has higher entropy, because matter has larger energy and hence access to a larger region of phase space. A dissipative phenomenon can move the system to a different trajectory, increasing the entropy and producing irreversibility. As we shall see in the next section, this is what happens in the universe when stars burn.

The energy exchanged between the single degree of freedom formed by the scale factor and all the others drive the full irreversibility of the universe we see. By far the dominant source of the irreversibility we observe is therefore the single fact that the scale factor a(t) was, in the technical sense above, out of equilibrium in the early universe (and it is still now). In the next section I am more precise about the cosmological evolution of entropy.

### 2.4. Going Out of Equilibrium Keeping Entropy Constant

The larger component of the entropy of the universe is the radiation. In the radiation dominated phase, the entropy of the radiation is
(4)$$S∼VT^{3}∼a^{3}a^{-3}∼constant,$$
where *V* is the volume and *T* is the temperature of a co-moving region, and is therefore constant. The irreversibility we concretely observe in the universe is due to phenomena that involve far smaller amounts of entropy than the large amount of entropy stored in the radiation, and are usually neglected in a first approximation.

Furthermore, perhaps confusingly, a system can go out of equilibrium even if its entropy remains constant. To see this, consider a co-moving volume in an idealized universe containing only two species of matter. Call *V* the volume, *U* the total internal energy of the matter and ρ the relative density (the number of particles of the first species over the total number of particles). The entropy, generally, is a function of these macroscopic variables.
(5)S=S(V,U,ρ).

Say at some initial time the value of the macroscopic variables is $\left(V_{0},U_{0},\rho_{0}\right)$ and the entropy $S_{0}=S\left(V_{0},U_{0},\rho_{0}\right)$. Consider the expansion to a state (V,U,ρ) with $V>V_{0}$. Because of homogeneity, there is no exchange of energy between co-moving regions, but matter exchanges energy with the gravitational field, therefore in general $U\neq U_{0}$. Since because of the homogeneity there is no exchange of heat either, as long as the expansion is reversible, entropy *S* remains constant:(6)$$S\left(V,U,\rho \right)=S_{0}.$$

In the course of an expansion from the volume $V_{0}$ to a volume *V*, there are two relevant possibilities: either the two species do not interact at all, each expanding freely, or they can be transformed into one another by some process, and ρ changes accordingly.

In the first case the ratio of the densities remains constant, $\rho =\rho_{0}$, and the change of *U* can be computed from the last equation. That is, the final value $U_{free}$ of the internal energy is determined by
(7)$$S\left(V,U_{free},\rho_{0}\right)=S_{0}.$$

In the second case, namely if the two species interact, transforming into each other, the density changes adjusting ρ to the value $\rho_{eq}\left(U,V\right)$ that maximizes the entropy at given *U* and *V*.
(8)$${\left.\frac{\partial S(V,U,\rho )}{\partial \rho }\right|}_{\rho =\rho_{eq}\left(U,V\right)}=0.$$

In this case the entropy is only a function of two variables,
(9)$$S\left(V,U\right)=S(V,U,\rho_{eq}\left(U,V\right)),$$
and the change of the energy is determined by this function remaining constant. That is, if the species interact, the final energy $U_{int}$ is determined by
(10)$$S\left(V,U_{int}\right)=S\left(V,U_{int},\rho_{eq}\left(U_{int},V\right)\right)=S_{0}.$$

In general $U_{free}\neq U_{int}$, because the work that the gravitational field does on the matter in the expansion from $V_{0}$ to *V* depends on whether there is interaction between the species or not: in the first case it must also account for energy needed to transform one species in the other.

Hence the two evolutions take the system to two states with the same volume and the same entropy, but different energy.

Imagine now that after an evolution where the species were not interacting (Hydrogen and Helium) a channel opens that allows one species to transform into the other (a star). Matter internal energy is conserved during this equilibration therefore the state $\left(V,U_{free},\rho_{1}\right)$ evolves to the state $\left(V,U_{free},\rho_{eq}\left(V,U_{free}\right)\right)$. This is different from the state $\left(V,U_{int},\rho_{eq}\left(V,U_{int}\right)\right)$ because $U_{free}$, as we have seen, is different from $U_{int}$, hence its entropy is different from $S_{0}$. It is higher, because it comes from a transformation towards equilibrium. Therefore there is a net increase of entropy
(11)$$\begin{array}{ccc} \Delta S & = & S\left(V,U_{free}\right)-S\left(V,U_{int}\right)\\ & = & S\left(V,U_{free}\right)-S_{0}. \end{array}$$
where $U_{free}$ is defined in (Equation 7) and S(U,V) in (Equation 9), see Figure 3.

This result may seem strange, because the system goes out of equilibrium even if its entropy remains constant. But of course there is nothing wrong with this; if we slowly shift a wall dividing a box filled with an ideal gas into two parts, the pressure dis-equilibrates, and—in this sense—we bring the system out of equilibrium without changing its entropy. We simply add energy bringing the system to a state where its maximal entropy could be higher; opening a hole in the wall starts an irreversible process. Stars are like that opening in the wall.

In conclusion, the universe has gone out of equilibrium at the freeze-out temperature. The overall entropy of matter and radiation has remained essentially constant, but the subsequent increase in volume has not been accompanied by the transfer of energy between gravity and the energy difference between hydrogen and helium that true matter equilibrium would have required, because the proton-neutron (and then hydrogen-helium) thermalization times have become too long, below the freeze-out temperature. Therefore, althout entropy remained constant, free energy has remained trapped into hydrogen. This is the free energy liberated in stars, which fuels the bulk of the irreversibility we witness.

Since the scale factor is a dynamical variable interacting with the other variables and since before the freeze-out time the radiation and matter content of the universe were in equilibrium among themselves, it is clear that it is the scale factor and only the scale factor that was not at equilibrium, and therefore was the ultimate source of low entropy.

### 2.5. Why Other Suspects Are Innocent

The scale factor a(t) is a component of the gravitational field. It is one of the dynamical variables of gravity. Therefore the above account shows that gravity has played a key role in the thermal history of the universe, via a(t).

A second role of gravity in the thermal history of the universe mentioned above is in the collapse of the hydrogen clouds that ignites stars.

Roger Penrose has emphasized a different role of gravity for past low entropy, and this idea has had a strong impact on the theoretical physics community, especially on the gravitational community. In the standard cosmological model, the gravitational field is very close to homogeneity and isotropy in the early universe. Penrose pointed out that this is an “extremely special” configuration of the gravitational fields, because a “generic” configuration of the geometry is strongly crumpled, not homogeneous [10]. This can be seen by studying the evolution of a re-collapsing universe: near the final big crunch, matter has collapsed into a large number of black holes and geometry is highly inhomogeneous. This progressive crumpling of the geometry can be seen as an entropy increase, from a low-entropy initial nearly spatially flat macro-state underpinned by a single microstate, towards a final higher entropy crumbled macro-state.

All this is of course theoretically correct, but it seems to me that there is clear evidence that this is not the source of the bulk of the irreversibility that we actually observe in our real universe. The reason is the following.

Imagine a universe—distinct from ours—where general relativity was not true, the metric of spacetime was precisely the metric (Equation 1) and the gravitational interaction was instantaneous and governed by Newton’s law of gravity. In this universe, the gravitational field would have no independent degrees of freedom at all, except for the scale factor. As fas as we understand, such a universe could be very similar to ours, with nearly the same thermal history. The reason being that this model does indeed account entirely for the current cosmological standard model. In fact, gravitational waves, metric perturbations of a Friedman cosmology other than the scalar potential, and black holes, play a small role in the thermal history of our universe: if we ideally shut down the independent degrees of freedom of gravity other than the scale factor, we still have virtually the same cosmological evolution and hence the same irreversibility. This observation implies that the irreversibility we see is not the consequence of the existence of these other gravitational degrees of freedom, because their absence would not avoid it.

I am not disputing Penrose’s observation that the initial state of spacetime is peculiar in the phase space of general relativity. But it seems clear that this peculiarity is not the main responsible for the irreversibility we actually see around us. The irreversibility we see is pretty much unrelated to the behaviour of gravitational radiative modes, and follows largely from one single degree of freedom, the scale factor, being far from an equilibrium configuration. If the existing black holes accounted for a considerable portion of the entropy of the universe, which is possible, I do not see how real black holes would substantially affect the irreversibility of our experience: we (and the galaxies) do not grow old because of black holes; we grow old because the thermal processes we are are fuelled by the free energy of the stars, which derives from the free energy in the primordial hydrogen.

Gravity influences the thermal history of the universe also by giving rise to the clumping of matter that ultimately leads to the rich structure of the universe. A detailed calculation (see for instance [6]), shows that a cloud itself lowers its entropy by shrinking, but emits heat, which rises the external (and the global) entropy. This suggests that we can relate the initial low entropy fuelling present irreversibility to the initial uniformity of the distribution of matter, whose entropy can increase by gravitational clumping. However, the total entropy produced by a star burning in enormously larger than the entropy produced by the heat emitted by the contraction of the hydrogen clouds. Hence once again, although matter uniformity might marginally contribute to current irreversibility, this effect is negligible with respect to the main one; the irreversibility produced by the fast increase of the scale factor, that have left hydrogen and helium badly trapped out of thermal equilibrium. The universe, in other words, has gone badly out of equilibrium much before any significative beginning of clumping of matter.

I think we can safely conclude that the past low entropy that gives rise to observed irreversibility is very largely in the single degree of freedom a(t) being not at an equilibrium value.

## 3. Part II: Why?

In this second part of the article, I discuss how to interpret the fact that the entropy of the early universe was low. To star with, let me review our present best understanding of the origin of the second law of thermodynamics. (See for instance [2] or the general introductory parts in [3,4]). This allows me to sharpen the question.

It is easy to make sense of the ubiquitous phenomenon of irreversibility of the observed processes in nature in statistical terms, if the initial state of these processes has low entropy. Indeed, the vast majority of the micro-states that underpin a low-entropy macro-state evolve towards a microstate underpinning a higher entropy configuration. Therefore, barring extraordinarily atypical micro-states, entropy grows if the initial macro-state of these processes has low entropy.

Given any specific irreversible process, its initial low entropy can be understood because of the way the process was prepared, by us or by Nature. In either case, preparation of an initial low-entropy state requires that the previous processes giving rise to the preparation themselves started of from an even lower entropy.

Tracing back in time, we arrive at the low entropy of the early universe. Therefore current irreversibility depends on early low-entropy in the universe. The open question about the arrow of time, therefore, is not why entropy grows: it is why entropy was low to start with.

This story implies that there is something ironic in the state of our understanding of irreversibility: the statistical understanding of entropy growth is grounded on the idea of genericity (generic micro-states underpinning low entropy states evolve towards higher entropy), but to make use of this idea we need to deal with the fact that the initial state had very low entropy, and very low entropy means to be very badly non-generic. Thus, our current understanding of irreversibility is based on an assumption of genericity (the micro-states underpinning macro-states are typical) and an assumption of non-genericity (initial entropy was very low). Can we make sense of this?

A possible attitude towards this question is to discard it. After all, “why” questions need to stop at some point. This is an attitude defended for instance by David Albert, who recommends to promote the “past hypothesis”, namely the statement that entropy was low in the past, to a sort of law of nature, in the sense of being a general statement from which we can derive predictions that turn out to be true [3]. To some extent, the discussion above supports David Albert’s position. Having identified initial low entropy with a small value of the scale factor, allows us to say that all current irreversibility is driven by the initial smallness of the universe, and this is it, as far as the arrow of time is concerned.

As for any other general fact of nature, not asking further why’s is a sensible option. Unless (or until) one finds something better. Can we find something better?

One possible relevant fact is that the scale factor might not have an equilibrium state at all. Hence there might not be state with maximal entropy at all and the universe might have no choice but being in a trajectory of increasing entropy. This idea is being explored for instance in [11,12,13] and I shall not pursue this line here.

### 3.1. The Role of Corse Graining

Let us reconsider the model of the gas coupled to the piston, and assume for simplicity that the total energy is bounded. Consider the space M of all possible motions of this system. This is a phase space (at any given time *t* it is in one-to-one correspondence with the space of the initial data) and carries a natural measure (the Liouville measure dμ, normalized by ∫dμ=1, which does not depend on *t*). For the vast majority of the motions in M, the piston is generally near its equilibrium value $X_{o}$:(12)$$\int_{M}X\left(t\right)\;d\mu ∼X_{o}.$$

Therefore if we witness a motion where the piston is far away from this value at some time $t_{0}$, we are witnessing a very atypical motion. For the same reason, we may say that we are witnessing a very “atypical” motion of our universe, among the motions allowed by its dynamics.

However, there is something missing in this account of what “atypical” means. The reason is that any individual motion of a system with many degrees of freedom is atypical in this sense. In fact, pick an arbitrary motion in M. A generic degree of freedom $x^{\left(n\right)}$ of the system has mean value
(13)$$\int_{M}x^{\left(n\right)}\;\left(t\right)\;d\mu =x_{o}^{\left(n\right)},$$
but out of many degrees of freedom there will generically most likely be some, say for instance *x*, such that $x(t_{0})$ is very different from $x_{o}$. Therefore in a system with many degrees of freedom there will generically be some degrees of freedom that are far out of equilibrium. So, having a single degree of freedom that is far out of equilibrium is not “atypical”: to the contrary: it is a generic occurrence: it is very typical! In which sense is then a low-entropy state “atypical”? What is missing?

What is missing is the fact that the notion of “typical” or “atypical”, like the notion of “low entropy” or “high entropy” make sense only if we patch together motions in families that we deem indistinguishable, namely if we “coarse grain” the phase space. Typicality makes no sense for micro-states alone: it only makes sense if macro-states are defined. Although rarely emphasized, it is essential to recall that irreversibility is a macroscopic notion. It is a property of a certain coarse graining. It is not a property of a microscopic dynamical evolution. In Appendix A I recall and discuss this essential point.

A macro-state is a defined as a subset *m* of M, or more in general, by a (normalized) distribution function ρ on M. Then the “typicality” of the macro-state is measured by the size of *m*
(14)$$S_{q}=\log\int_{m}d\mu$$
or the entropy of ρ
(15)$$S=-\int_{M}\rho \log\rho .$$

In particular, macro-states are defined if something selects some variables of the system as “macroscopic”. For instance, in the second model considered above, the volume of the cylinder (namely the position of the piston) and the total energy *E* were declared “macroscopic” variables. They define the macro-states formed by the region $M_{V,E}$ in M where the volume is *V* or smaller and the energy equal or lower than *E*. The entropy of this region $S_{V,E}$ is a well defined quantity.

These observations show that the reason why the scale factor determines an initial low entropy is not just because it is at a value which is not at equilibrium: rather it is because the scale factor is considered a macroscopic variable. This is the main conceptual point I wish to make in this paper. Past low entropy is due to the fact that a single variable a(t) that happened to be far from its equilibrium value in the early universe is a variable that we treat as “macroscopic”. Why do we treat it as macroscopic?

### 3.2. Why We Treat the Scale Factor as Macroscopic?

What is the difference between a “macroscopic” and a “microscopic” vraiable? There are different answers to this question that one can find in the literature. These are:External interactions. The classical paradigm of a thermodynamical system is a physical system with many degrees of freedom $x_{n}$, acted upon by an agent that can control and measure a small number of its variables, $X_{n}$ (the thermodynamical variables). The $X_{n}$’s are the macroscopic variables that determine the statistical coarse graining, and underpin the definition of entropy in the statistical analysis of the system. The coarse graining is not arbitrary: it is physically determined by the interactions of the system with the agent. Thermodynamics describes the macroscopic behaviour of systems relative to the sets of existing physical interactions with an external agent measuring it and acting on it. When dealing with the thermal history of the entire universe, it is hard to apply this characterization of “macroscopic”: there is no “external” agent acting on the universe.Heat versus work. Thermodynamics has developed as the science describing the exchanges of heat and work between a system and its environment. What is it that determines the difference between heat and work? Work is a form of mechanical energy. But so is heat, at the microscopic level. When we give a macroscopic account of a process, heat is not considered mechanical energy anymore only because the relevant degrees of freedom are not directly accessible. The distinction between heat and work is therefore more subtle than what it seems at first sight: heat refers to energy in microscopic variables, while work refers to energy in macroscopic variables. The distinction between heat and work, therefore does not determine the characterization of what is macroscopic: it follows from it.Averages. Boltzmann’s approach to statistical mechanics considers a system *S* formed by a large number *N* identical copies $s_{n}$ of a simple system *s*. The typical example is a gas formed by many similar molecules. We can then define a distribution $\rho :\sigma \to R^{+}$ on the phase space σ of *s*, which assigns to any region R⊂σ the fraction of the molecules that are in states in this region. That is, if the *n*-th molecule is in the state $x_{n}\in \sigma_{n}$,
(16)$$\int_{R\subset \sigma }\rho =\frac{1}{N}\sum_{n}\int_{R\subset \sigma_{n}}\delta_{x_{n}}.$$
Then an observables *o* of a single molecule defines a macroscopic observable *O* for the full system, by its average under this distribution:
(17)$$O=\int_{\sigma }o\rho .$$
This is a powerful tool that exploits the fact that the system is formed by many copies of a single system. However, it can be applied only for systems composed by a large number of identical subsystems. This is not always the case with field theory, general relativity, or the entire universe.Relative entropy. Point (i) above can be stripped of its anthropocentric and subjectivist aspects as follows. If a system *S* with many degrees of freedom $x_{n}$, interacts with another system *O* via an interaction hamiltonian that depends on a small number of variables $X_{n}$ of *S*, then a coarse graining on *S* is defined by defining the variables $X_{n}$ as macroscopic. This defines an entropy for *S*, which is objective but relative to *O*. This definition of “macroscopic” can be extended to the case of a closed system like the universe by identifying *O* as a subsystem of the universe and *S* as “the rest” of the universe.

At the light of this list, why is the scale factor a(t) a macroscopic variable?

The only definition that applies, seems to me, is the last one: relative entropy. We are part of the universe. Hence we belong to some subsystem *O* of the universe. The subsystem *O* concretely interacts with the rest of the universe via variables $X_{n}$ whose number is small compared to the number of degrees of freedom of the universe. These are the macroscopic variables in a statistical description of he universe. They define a coarse graining, hence an entropy. The scale factor a(t) belongs to this set $X_{n}$, otherwise we could not do cosmology.

The conclusion is that past low entropy depends on the fact among the relatively few macroscopic variables that determine our own interaction with the rest of the universe, there is one that was out of equilibrium in the past.

This discussion opens a possible way to interpret the atypicality implicit in past low entropy. What is atypical is not something pertaining to the universe by itself, but to the interacting couple (S,O). This opens the possibility that what is atypical is *O*, not the state of *S*. This is the idea of the perspectival origin of the arrow of time that was put forward in [7]. It is based on a simple conjecture in statistical mechanics:

Conjecture: In a sufficiently complex dynamical system *S* with sufficiently many interacting degrees of freedom $x_{n}$, for any generic finite motion there are some subsystems *O* that interact with the rest of *S* via interaction variables $X_{n}$ that define a coarse graining and hence an entropy for the rest of *S* that is arbitrarily low at one extreme of the motion.

If this conjecture is true, as it seems intuitively possible, then the universe was not in a a-typical state in the early universe. It is us that we happen to belong to one of the subsystems that the conjecture states exist generically. (On this, see also [14]. The precise relation with the argument in this reference will be studied elsewhere.) The reason we happen to be part of one of these peculiar systems is simply that these are the systems constructed in terms of those macroscopic variables for which entropy was low in the past and hence increases in time. We are the product of this perspectival growth of entropy. When seeing the arrow of time, we are not seeing a property of the microscopic motion of the universe: we are seeing a feature of those special macroscopic variables that characterize us.

If this is the case, the arrow of time is real, but it is perspectival, like are real but perspectival the rotation of the sky or the setting of the sun.

## 4. Conclusions

In the first part of this note I have argued that the dominant source of the low-entropy of the past universe is only the smallness of the scale factor, which is far from an equilibrium value. Because of this lack of equilibrium, the rapid expansion of the universe creates metastable states (hydrogen not fused to helium) and then dissipation, and irreversibility.

In the second part of the note I have discussed the extent to which this fact requires an assumption of non-typicality. I have observed that the low entropy is not due to the fact that one variable is far from equilibrium, but rather to the fact that one of the variable very far from an equilibrium configuration is a variable that we treat as macroscopic.

I have then observed that the reason for which we consider this variable macroscopic is not completely clear. I see two possibilities. Either the variable is objectively “special”, or it is special because it belongs to the macroscopic variables determined by the peculiar interaction between a physical subsystem to which we belong and the rest of the universe.

This second possibility opens up the speculative possibility that the arrow of time is perspectival: if a plausible conjecture on statistical mechanics hold, what may be special is not the state of the universe, but rather the set of macroscopic variables we use to describe the macroscopic universe. The arrow of time might be real, but perspectival, like the rotation of the sky around us, as argued in [7].

## Figures and Tables

**Figure 1 entropy-21-00466-f001:**
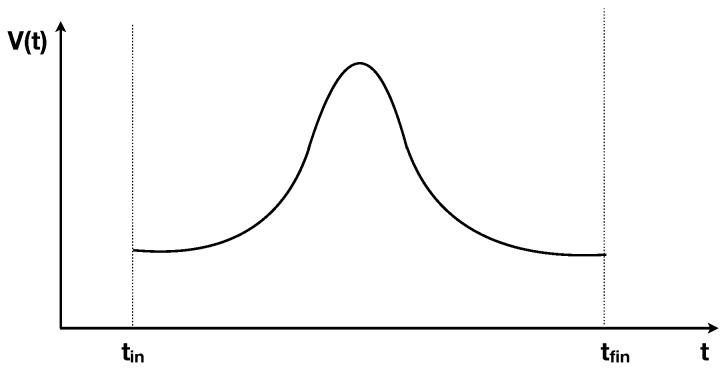
Time evolution of the volume in the gas model.

**Figure 2 entropy-21-00466-f002:**
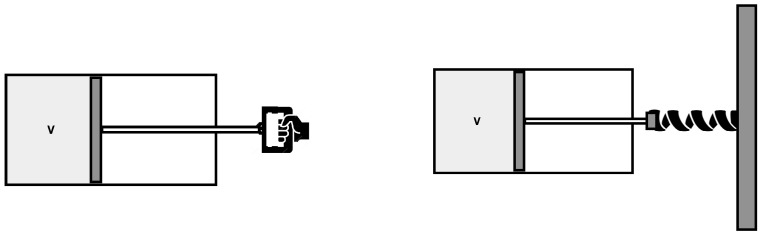
The two models considered in the paper. (**left**) A system formed by the gas in a chamber; its volume is changed by a piston moved by an external force. (**right**) A system formed by a gas in a chamber plus the piston attached to a spring.

**Figure 3 entropy-21-00466-f003:**
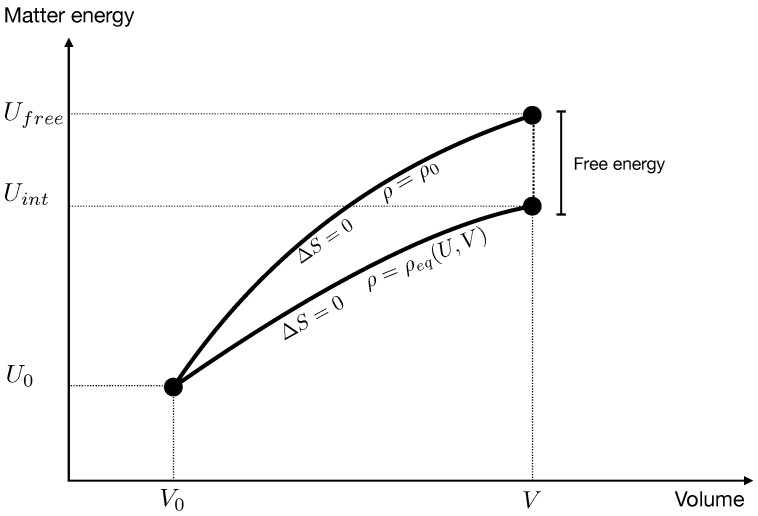
The expansion of a co-moving region from a a volume $V_{0}$ to a volume *V* without or with interaction between two matter species. In the first case the relative density ρ remains constant; in the second, the density adjusts itself to maximize entropy. The end point of the first evolution is metastable, the end point of the second is stable. In both evolutions the entropy is constant, but in the first matter extracts more energy from the gravitational field. This energy is free energy that can fuel irreversibility if a channel of interaction between the two species is open, as happens in stars. The last step is the dissipative equilibration process.

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
