# Peer review of "Where Was Past Low-Entropy?"

_entropy, 2019, doi:10.3390/e21050466_

Round 1
Reviewer 1 Report
I like the paper for its attempt at pedagogic clarity. However there are some
important conceptual problems that need to be addressed before publication.
1) the abstract needs a conclusion. Where precisely is the low entropy located in the early universe? The abstract needs to answer that question….not just say that you answer it.
2) In Kolb and Turner and in standard cosmology, the entropy of the CMB
(or any relativistic particles) is isentropic because S ~ VT^3 ~ a^3 (a^-3).
Since the very early universe is radiation dominated, the expansion is isentropic and the arguments given for entropy increase in this paper are only valid (if at all) in the matter dominated part of cosmic evolution. The behaviour of a(t) depends on whether we are in the radiation or matter-dominated epoch. This issue needs to be addressed..
3) Assuming there is some equilibrium value of a(t) between 0 and infinity seems ad hoc.
I don’t see any justification for it, and yet the whole argument depends on it.
There are also many typos that need fixing.
Author Response
1) The abstract needs a conclusion. Where precisely is the low entropy located in the early universe? The abstract needs to answer that question….not just say that you answer it.
I have modified the abstract as suggested. Thanks.
2) In Kolb and Turner and in standard cosmology, the entropy of the CMB (or any relativistic particles) is isentropic because S ~ VT^3 ~ a^3 (a^-3). Since the very early universe is radiation dominated, the expansion is isentropic and the arguments given for entropy increase in this paper are only valid (if at all) in the matter dominated part of cosmic evolution. The behaviour of a(t) depends on whether we are in the radiation or matter-dominated epoch. This issue needs to be addressed..
The referee is right of course. This gives me the opportunity to clarify this point. I have rewritten the paragraph where I show that a system can go out of equilibrium also in a process in which the entropy remains constant.
3) Assuming there is some equilibrium value of a(t) between 0 and infinity seems ad hoc. I don’t see any justification for it, and yet the whole argument depends on it.
Thanks for pointing this out. In fact, the first version of the paper did give the impression that the argument depend on the existence of an equilibrium value for a(t). It does not. I have clarified this point, adding a paragraph to explain it and correcting the text where there was a reference to the equilibrium value of a(t). The only point needed for the argument is that a(t) was not at an equilibrium, whether or not an equilibrium value exist.
4) There are also many typos that need fixing.
Thanks. I have corrected many of them.
Reviewer 2 Report
In this paper the author addresses the issue of the initial low entropy of the university by using some classical thermodynamical concepts. The paper contains some interesting ideas they are quite limited. It shows some (extremely simplified) thermodynamical considerations (and without a detailed discussion about the concept of entropy that plays a key role) only by explaining some toy models and by doing rough analogy with cosmology of the early universe. The paper could be accepted for publication after some revision and some clarifications. I would like the author to address several issues:
- First of all, there are a lot of typos in the paper that sometimes difficult the reading
- In the title, abstract and first line in the introduction it is claimed that it is identified exactly where the past low entropy is located. I think that is not true and it overrates what is done in the paper, so it should be softener
- In the first paragraph of the introduction, the sentence “The conclusions of this first part seem solid to me” is out of physical context and should be modified
- Last paragraph of section 2.2: Where it is said “This is what happened in our universe”. That sentence should be softener and explain the possibility is exposed here. In addition, it is said that there is only one degree of freedom (the scale factor). That simplified consideration should be explained.
- Section 2.3 as it is written it is not explaining any phenomena. It is only describing the low entropy by asuming that the scale factor is out of equilibrium, but not giving reasons for it or describing the system the author is considering.
- In section 2.4 I do not see clear the statement of considering the maximal entropy determined only by adjusting the change in the density and what is the equilibration process the author is considering.
- In section 2.5, first of all I would appreciate some clarification about the gravitational interaction the author is considering for the analogy that is “instantaneous and Newtonian and has no local degrees of freedom except for the scale factor”. Without any further clarification there is not possible to make a clear analogy as the author is doing, claiming a similar thermal history of universes with different gravitational laws.
- In section 3.1, regarding the consideration of macrostates and the role of coarse-graining, I do not see the claim of low entropy due to a single variable being macroscopic, that would be relevant for the conclusions. I find in general some confusion between the concepts macroscopic and microscopic and a coarse-grained variable.
- In section 3.2, I miss some discussion about the observer dependence of entropy, apart from the indirect comments in question (1) and the last paragraph of the section. As they are now, it shows an incomplete picture.
- In section 3.2, as well I find the definition of heat and work in terms of consider them macroscopic or not a little bit poor and that should be addressed following something similar to my previous two comments.
- In the end of section 3.2 the statement of the entropy gradient should be explained. Otherwise, the explanation he presents is not better than the issue he is addressing in the paper. Is this the final explanation of the past low-entropy?
- In appendix A, the author comment again about the scale factor being out of equilibrium. As I commented before, that assumption should be clarified.
- In appendix B, in the second paragraph it is not clear how the author justifies what it is said in the rest of the section. That is, is there irreversibility in microphysics or not? Where it appears and which limitations has it?
Finally, the last paragraph of appendix B, underlies something similar to some of my previous questions: is it observer dependent? What kind of coarse-graining is considered?
Author Response
- First of all, there are a lot of typos in the paper that sometimes difficult the reading.
Thanks. I have corrected many of them.
- In the title, abstract and first line in the introduction it is claimed that it is identified exactly where the past low entropy is located. I think that is not true and it overrates what is done in the paper, so it should be softener
Thanks, I have softened the claim and rewritten the abstract.
- In the first paragraph of the introduction, the sentence “The conclusions of this first part seem solid to me” is out of physical context and should be modified
I have dropped it.
- Last paragraph of section 2.2: Where it is said “This is what happened in our universe”. That sentence should be softener and explain the possibility is exposed here. In addition, it is said that there is only one degree of freedom (the scale factor). That simplified consideration should be explained.
I have completely changed this statement, which was indeed excessive. Thanks.
- Section 2.3 as it is written it is not explaining any phenomena. It is only describing the low entropy by asuming that the scale factor is out of equilibrium, but not giving reasons for it or describing the system the author is considering.
I have added a full section explaining this point. Last two paragraphs of Section 2.3.
- In section 2.4 I do not see clear the statement of considering the maximal entropy determined only by adjusting the change in the density and what is the equilibration process the author is considering.
I have rewritten the section making it more clear and answering the question. The equilibration process is the opening of the channel (for instance in stars) that allows hydrogen to fuse into helium, hence moving form the metastable state to a more stable one.
- In section 2.5, first of all I would appreciate some clarification about the gravitational interaction the author is considering for the analogy that is “instantaneous and Newtonian and has no local degrees of freedom except for the scale factor”. Without any further clarification there is not possible to make a clear analogy as the author is doing, claiming a similar thermal history of universes with different gravitational laws.
I have improved this part of the paper, making it more explicit.
- In section 3.1, regarding the consideration of macrostates and the role of coarse-graining, I do not see the claim of low entropy due to a single variable being macroscopic, that would be relevant for the conclusions. I find in general some confusion between the concepts macroscopic and microscopic and a coarse-grained variable.
I have rewritten this section more clearly.
- In section 3.2, I miss some discussion about the observer dependence of entropy, apart from the indirect comments in question (1) and the last paragraph of the section. As they are now, it shows an incomplete picture. - In section 3.2, as well I find the definition of heat and work in terms of consider them macroscopic or not a little bit poor and that should be addressed following something similar to my previous two comments. - In the end of section 3.2 the statement of the entropy gradient should be explained. Otherwise, the explanation he presents is not better than the issue he is addressing in the paper. Is this the final explanation of the past low-entropy?
I have rewritten Section 3.2 more clearly. One final possible explanation considered is that past low entropy is perspectival. That is, it is due the particular manner the physical system we belong to interact with the rest of the universe.
- In appendix A, the author comment again about the scale factor being out of equilibrium. As I commented before, that assumption should be clarified.
I have dropped Appendix A, which was more misleading than clarifying.
In appendix B, in the second paragraph it is not clear how the author justifies what it is said in the rest of the section. That is, is there irreversibility in microphysics or not? Where it appears and which limitations has it?
I have rewritten appendix B focusing it only on the fact that irreversibility is a macroscopic phenomenon only defined if a coarse graining is given. This avoid confusing repetitions with the main text.
Round 2
Reviewer 2 Report
The author replied all my comments and he improved the paper by explaining the mentioned issues. I consider that now the paper is enough interesting to be published in the journal.